# Broadband miniaturized spectrometers with a van der Waals tunnel diode

Md Gius Uddin [1,2], Susobhan Das[1], Abde Mayeen Shafi[1], Lei Wang[3], Xiaoqi Cui [1,2], Fedor Nigmatulin[1,2], Faisal Ahmed[1], Andreas C. Liapis [2], Weiwei Cai[3], Zongyin Yang [4], Harri Lipsanen [1], Tawfique Hasan [5], Hoon Hahn Yoon [1,6] & Zhipei Sun [1,2] ✉

Miniaturized spectrometers are of immense interest for various on-chip and implantable photonic and optoelectronic applications. State-of-the-art conventional spectrometer designs rely heavily on bulky dispersive components (such as gratings, photodetector arrays, and interferometric optics) to capture different input spectral components that increase their integration complexity. Here, we report a high-performance broadband spectrometer based on a simple and compact van der Waals heterostructure diode, leveraging a careful selection of active van der Waals materials- molybdenum disulfide and black phosphorus, their electrically tunable photoresponse, and advanced computational algorithms for spectral reconstruction. We achieve remarkably high peak wavelength accuracy of ~2 nanometers, and broad operation bandwidth spanning from ~500 to 1600 nanometers in a device with a ~ 30×20 μm² footprint. This diode-based spectrometer scheme with broadband operation offers an attractive pathway for various applications, such as sensing, surveillance and spectral imaging.

Optical spectrometers, which analyze the spectral contents of light, are cornerstone instruments in a wide variety of application fields ranging from fundamental scientific research to industrial inspections[1,2]. Conventional spectrometers are bulky and costly as they typically need several movable dispersive optical components (e.g., motorized gratings, interferometers) and thousands of detectors or filter arrays. Although these benchtop spectrometers offer high resolution and wide spectral range, their large physical dimensions restrict them from being widely adopted in numerous portable applications, such as consumer electronics, smart wearable devices, drone integration and remote sensors[3].

Over the past few years, various approaches have been developed to realize miniaturized spectrometers with a significant reduction in

their cost and footprint by replacing the bulk dispersive optical components with tunable filter arrays or compact interferometers[4,5]. These techniques offer excellent performance in general, but they cannot be scaled down below the sub-millimetre scale due to fundamental physical limitations in the optical path length. To overcome these restrictions, a "reconstructive-type" operation principle has recently been developed that relies on special computational algorithms for spectral reconstruction[6,7]. Examples of such approaches include spectrometers realized on colloidal quantum dots[8], in-situ dynamically modulated perovskites[9], compositionally engineered[10] and superconducting[11] nanowires. Two-dimensional (2D) materials[12,13] and their heterostructures[14,15] have been used to fabricate ultra-compact computational spectrometers owing to their inherently

¹Department of Electronics and Nanoengineering, Aalto University, Espoo 02150, Finland. ²QTF Centre of Excellence, Aalto University, Aalto 00076, Finland. ³Key Lab of Education Ministry for Power Machinery and Engineering, School of Mechanical Engineering, Shanghai Jiao Tong University, Shanghai 200240, China. ⁴College of Information Science and Electronic Engineering and State Key Laboratory of Modern Optical Instrumentation, Zhejiang University, Hangzhou 310027, China. ⁵Cambridge Graphene Centre, University of Cambridge, Cambridge CB3 0FA, UK. ⁶Department of Semiconductor Engineering, School of Electrical Engineering and Computer Science, Gwangju Institute of Science and Technology, 123 Cheomdangwagi-ro, Buk-gu, Gwangju 61005, Republic of Korea. ✉e-mail: zhipei.sun@aalto.fi

dangling-bond-free surfaces, atomically sharp interfaces, layer-dependent bandgap, and electrically tunable photoresponse[16–18]. However, the operation bandwidth, precision, and portability of all these miniaturized spectrometers are generally restricted by the bandgap engineering limits of their chosen materials[19].

In this work, we report a reconstructive broadband spectrometer based on van der Waals heterostructures consisting of an overlapping tunnel junction of two semiconducting 2D materials—molybdenum disulfide ($MoS_2$) and black phosphorus (BP). In contrast to the previous 2D materials-based spectrometers[13–15], which typically use both gate-source and drain-source voltages to tune wavelength-dependent photoresponses, here, we propose and demonstrate to use only the drain-source voltage to create an electrically tunable spectral response for spectral reconstruction. In this design, the spectrometers are much more compact and simpler in architecture, as gate contacts and insulation layers are not required. Further, in contrast to the previously reported results (e.g., pure BP[12], graphene[13], $MoS_2/WSe_2$[14], $ReS_2/Au/WSe_2$[15]), we choose $BP/MoS_2$ heterostructures because the $BP/MoS_2$ junctions exhibit the transition of band alignment between staggered- and broken-gap. This can effectively switch the charge carrier transport from thermionic emission to band-to-band tunneling only by tuning the source-to-drain bias voltage across the junction without gating[20–24]. This drastically reduces the contact resistance under the reverse bias voltage and enables low-voltage operation without a gate terminal. Our new van der Waals tunneling-diode-driven spectrometer has a broad operation bandwidth from the visible to near-infrared (NIR) ranges. Additionally, the demonstrated small operation voltage change (-2 V) of the diode-based spectrometer design offers immense potential in realizing practical devices for various on-chip applications.

## Results

### Operation principle

Figure 1 illustrates the operation principle of our diode-based broadband spectrometer concept. It involves three distinct steps: (a) learning, (b) testing, and (c) reconstruction. The optoelectronic properties of a diode can be tuned in a controlled manner by applying a tunable external electric bias potential across the junction. As shown in Fig. 1a, the responses of the diode are first studied with multiple known monochromatic optical inputs individually. This leads to a distinct photoresponse matrix of the device. The corresponding photo-responsivity can be treated as a function of both the incident light wavelength ($\lambda$) and the external electric bias potential ($V_{ds}$) applied across the junction. The entire photoresponse information is encoded into a single responsivity matrix ($R$, Right panel of Fig. 1a) whose electrically tunable spectral response elements ($R_{ij}$) are denoted as-

$$R_{ij} = \frac{I_{ph}\left(\lambda_i, V_{dsj}\right)}{P} \ldots \quad (1)$$

where $I_{ph}$, $\lambda$, and $P$ represent the generated photocurrent, the wavelength of the incident light, and the optical power of the incident light, respectively. In the testing step, as shown in Fig. 1b, the bias-dependent photocurrent response of the device corresponding to the incident light with an unknown spectrum is measured. For signal processing, the incident light is assigned to an unknown spectrum function ($S(\lambda)$). The photoresponse data measured in the testing step, together with the calibrated response functions obtained in the learning step, are then processed to reconstruct ($S(\lambda)$) by solving a system of linear equations[10]:

$$\int_{\lambda_{min}}^{\lambda_{max}} S(\lambda).R_i(\lambda)d\lambda = I_i (i=1,2,3\ldots,n)\ldots \quad (2)$$

where $\lambda_{min}$ and $\lambda_{max}$ refer to the operational bandwidth of the spectrometer. We compute its constrained least-squares solution to reconstruct the unknown incident light spectrum (Fig. 1c).

### Device scheme and characterization

Figure 2a illustrates the schematic view of our $BP/MoS_2$ diode-based spectrometer fabricated on a Si substrate with a 300 nm thermally grown $SiO_2$ capping layer. Note that hexagonal boron nitride (hBN) passivation layers encapsulate the device. There is an additional $Al_2O_3$ encapsulation on top of the hBN. The device fabrication process is described in the Methods section. A corresponding optical microscope image of the spectrometer is provided in Fig. 2b, where multi-layer BP and $MoS_2$ flake edges are depicted with dashed yellow and

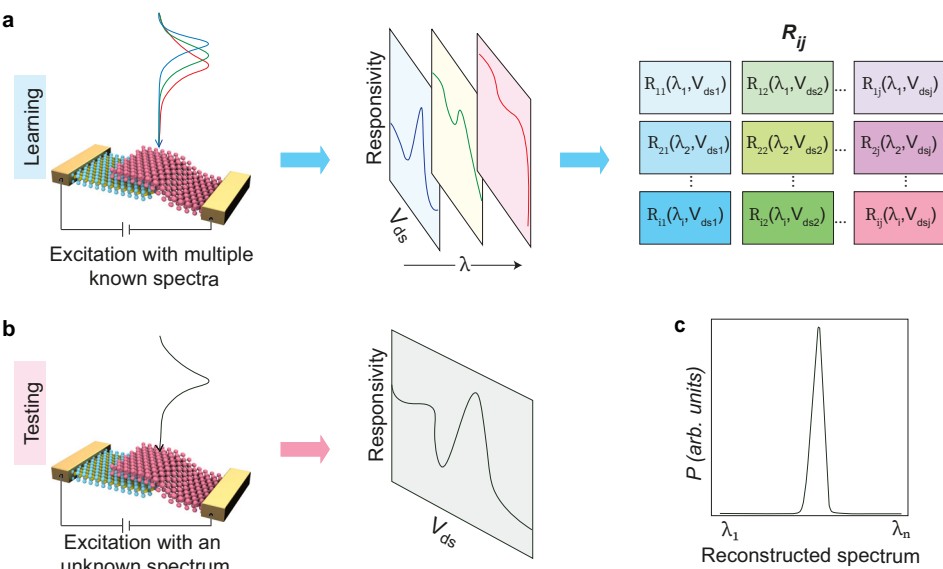

**Fig. 1 | Schematic of the working principle of our diode-based broadband miniaturized spectrometers. a** In the learning step, the diode is excited with multiple known monochromatic inputs (Left panel). The bias-dependent ($V_{ds}$) spectral responses (Middle panel) are encoded into responsivity matrix elements (Right panel). **b** In the testing process, the electrical response (Right panel) of the incident light with unknown spectral information is recorded (Left panel). **c** Spectral information of unknown incident light is then reconstructed using results obtained in the learning and testing steps with computational algorithms.

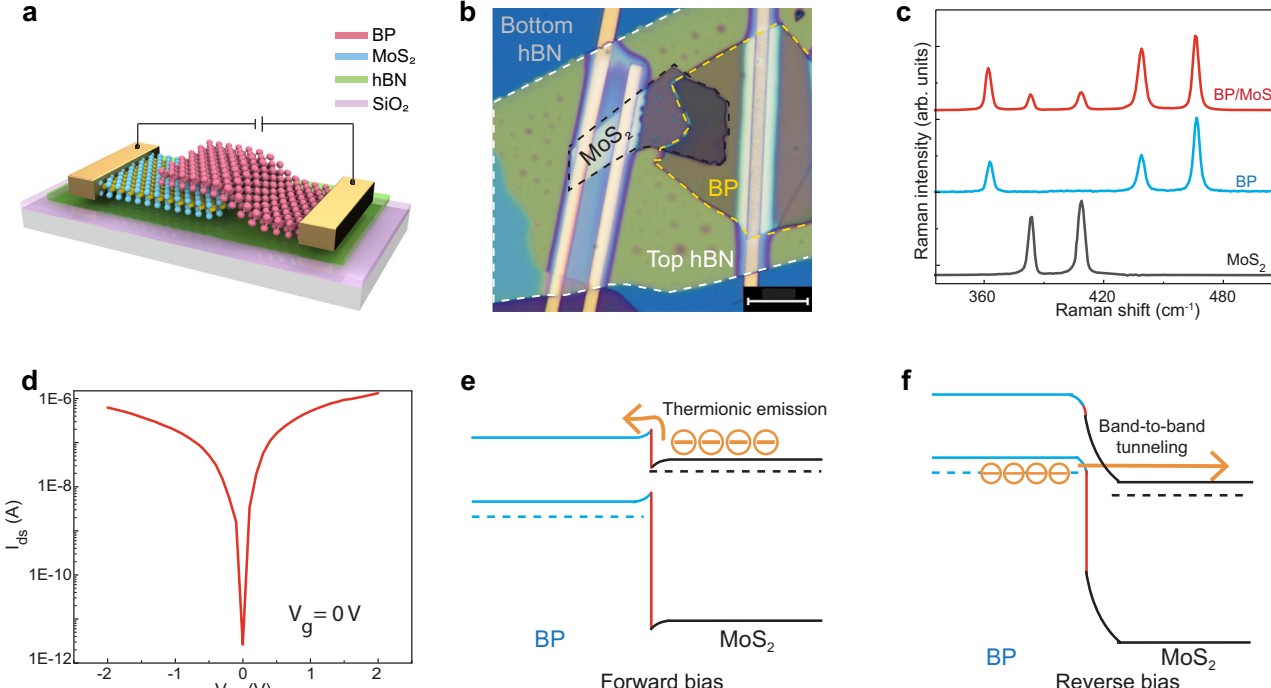

**Fig. 2 | Characterization of our broadband diode-based spectrometer.**
**a** Schematic of our BP/MoS$_2$ diode spectrometer. The junction is encapsulated by hBN passivation flakes. The top hBN and an additional Al$_2$O$_3$ protection layer are not shown. **b** Corresponding optical microscope image. The black and yellow dashed lines indicate MoS$_2$ and BP flakes, respectively. Dashed white line marks the edges of the top hBN flake. Scale bar: 10 μm. **c** Raman spectra of bare MoS$_2$, BP, and the BP/MoS$_2$ heterostructure. **d** Log plot of $I_{ds}$-$V_{ds}$ characteristics of the BP/MoS$_2$ diode. **e**, **f** Band diagram and carrier transport mechanisms across the BP/MoS$_2$ heterodiode under forward (**e**) and reverse (**f**) bias conditions.

black lines, respectively. We fabricate electrodes on both BP and MoS$_2$ flakes to analyze their electrical characteristics.

In this work, we use multilayer flakes to fabricate the BP/MoS$_2$ heterostructure to achieve high photoresponse. Atomic force microscopy (AFM) mapping on the heterostructure spectrometer presented in Fig. 2b reveals the thicknesses of the BP and the MoS$_2$ flakes as ~45 and ~33 nm, respectively (Supplementary Fig. S1). To confirm material quality, we perform Raman measurements on pure channels and overlapping regions of the devices. Figure 2c shows the Raman spectra of bare MoS$_2$ (black line), BP (blue line), and their heterostructure regions (red line) excited by a 532 nm laser at room temperature. Two distinct Raman modes of MoS$_2$ are observed in the range of ~330–510 cm$^{-1}$: $E_{2g}^1$ at ~383 cm$^{-1}$ and $A_{1g}$ at ~407 cm$^{-1}$; while BP shows three phonon modes: $A_g^1$ at ~362 cm$^{-1}$, $B_{2g}$ at ~439 cm$^{-1}$, and $A_g^2$ at ~467 cm$^{-1}$, in good agreement with previous reports[22,25]. Raman peaks of the heterostructure region comprise the contribution from both flakes, indicating the formation of a high-quality van der Waals heterojunction.

We perform electrical measurements on the tunnel diode to reveal its intrinsic properties. For the heterojunction, drain bias is applied to BP, while MoS$_2$ is connected to the source terminal. In this measurement configuration, electrons are injected from the source electrode to the MoS$_2$ flake and subsequently transported to the BP flake through the heterostructure region. Eventually, the electrons are collected by the drain electrode. Figure 2d presents the output characteristics of our BP/MoS$_2$ heterostructure on a logarithmic scale. The junction offers comparable conduction for forward and reverse $V_{ds}$. The superior current level under reverse bias is attributed to the band-to-band tunneling (BTBT) mechanism, as discussed later. Since we use multilayer BP and MoS$_2$ flakes, the quantum confinement effect can be ignored. Consequently, their band profiles are considered the same as their bulk counterparts. BP and MoS$_2$ have electronic bandgaps of ~0.3 and 1.29 eV, respectively. Multilayer BP typically behaves as a

degenerate p-type semiconductor whose Fermi level lies in the valence band, while multilayer MoS$_2$ shows normal n-type behavior (Supplementary Fig. S2). When a forward $V_{ds}$ is applied across the heterojunction, as shown in Fig. 2e, the band of BP shifts downward compared to its equilibrium state. Consequently, the interface barrier height for electrons decreases. Therefore, conduction band electrons (majority carriers) of MoS$_2$ effectively surmount the reduced interface barrier height, enabling forward conduction. This forward conduction is a thermionic emission process where the forward current increases exponentially with increasing $V_{ds}$ in the ideal case. Since BP is a strong p-type and MoS$_2$ is an n-type semiconducting material, their respective minority carrier concentration is intrinsically low, limiting their contribution to forward conduction. Figure 2f shows a heterostructure band diagram under reverse bias. The degenerated BP offers high doping concentration, and its electrons filled below the valence band level can tunnel to the allowed states above the conduction band level of MoS$_2$.

## Spectrometer demonstration
Upon optical irradiation of the heterojunction, the generation of photocurrent occurs depending on the signatures of excitonic states in the absorption spectrum of constituent flakes[26,27]. When the BP/MoS$_2$ diode is excited with light in the visible wavelength region, as shown in Fig. 3a, the electrons in the valence bands of both MoS$_2$ and BP can be excited to their respective conduction bands. Under forward bias (left panel of Fig. 3a), the photoexcited electrons in the conduction band of MoS$_2$ are injected into the conduction band of BP. Under reverse bias (right panel of Fig. 3a), the valance band electrons of BP tunnel to the conduction band of MoS$_2$ while the photoexcited electrons in the BP conduction band drift to the MoS$_2$ conduction band and result in a strong photocurrent signal.

In the learning process of the spectrometer, we first measure the output curves of the BP/MoS$_2$ heterojunction diode under multiple

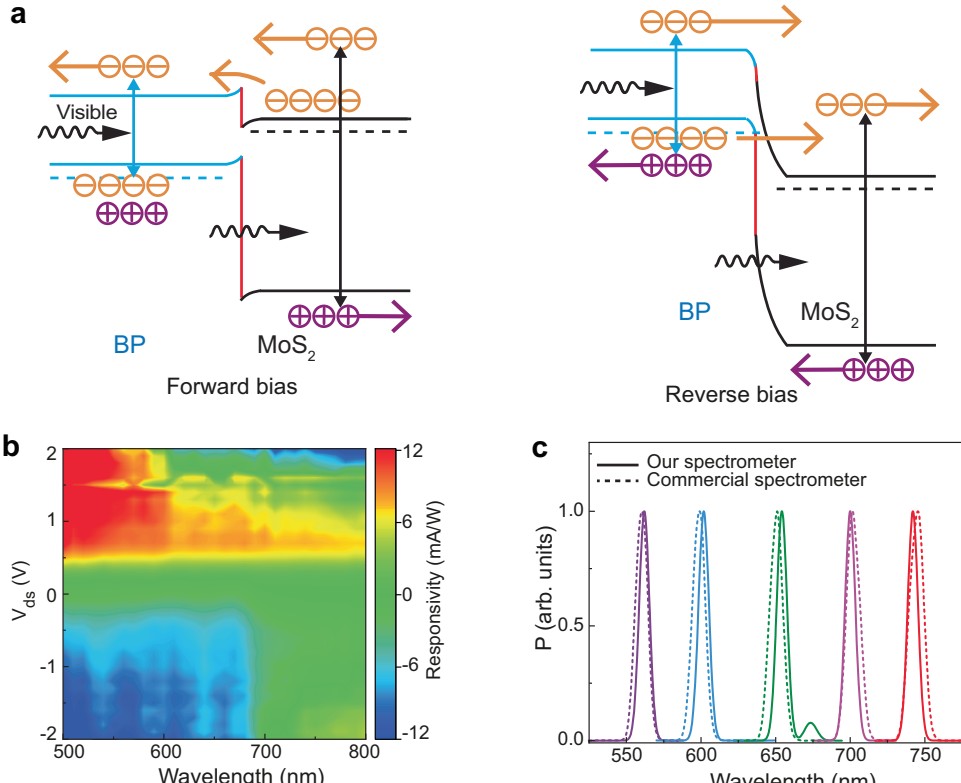

**Fig. 3 | Demonstration of our spectrometer at the visible range. a** Schematic of photocurrent generation mechanism when the junction is excited with the visible light under forward bias (left panel) and reverse bias (right panel) conditions. **b** Color contour plot of experimentally obtained spectral response matrix (31 different spectra, 41 voltage values per spectrum). **c** Quasi-monochromatic spectra reconstructed with our spectrometer (solid curves). Dashed curves represent corresponding spectra obtained with a commercial spectrometer.

known incident lights with a spectral width of ~10 nm. For the operation demonstration at the visible spectral region, we vary the incident excitation from ~500 nm to 800 nm in a step of 10 nm. Figure 3b shows the calculated photoresponsivity ($R$), also called responsivity in short, as a function of the wavelength of incident lights and the bias voltage applied across the heterojunction. The responsivity is defined as $R = I_{ph}/P$; where $I_{ph}$ and $P$ represent the photocurrent and the optical power of incident light, respectively. After encoding the spectral response matrix for the learning process, we measure unknown incident light spectra following the spectrum reconstruction process (Supplementary S3). In brief, we first measure the bias-tunable photocurrent of the unknown incident light (Fig. 1b). Afterwards, we compute its constrained least-squares solution to reconstruct the spectrum using an adaptive Tikhonov method[10,14]. Details of the optical setup, electrical, and optoelectrical measurements are provided in the Methods section. As shown in Fig. 3c, the quasi-monochromatic spectra reconstructed with our spectrometer agree with the reference spectra measured using a commercial spectrometer. The average peak-wavelength difference ($\Delta\lambda$) between reconstructed and reference spectra is calculated as ~2.5 ± 0.9 nm (Fig. 3c). The small second peak at ~675 nm is a computational error. In principle, such artifacts can be suppressed by filtering or improving the base model. We also carry out broadband measurement (Supplementary Fig. S4), which agrees well with that of the commercial spectrometer.

We now demonstrate the operation of our spectrometer at the NIR spectral region. Under NIR illumination at the heterojunction tunnel diode, as shown in Fig. 4a, the photocurrent generation process is mainly attributed to the direct-bandgap transition in BP since the incident photons cannot provide sufficient energy to excite the electrons in $MoS_2$ from its valence band-to the conduction band. Under reverse bias (right panel of Fig. 4a), the photoexcited electrons in the

BP drift to the $MoS_2$ conduction band. To demonstrate operation capability at the NIR region with high spectral resolution using our single-junction ultraminiaturized spectrometer, we construct a spectral response matrix by choosing a small (fine) learning step. Figure 4b presents the responsivity matrix obtained in the learning process for monochromatic incident lights ranging from ~1550 to 1560 nm in a step of ~0.5 nm. After learning, we test our spectrometer. The results shown in Fig. 4c indicate our broadband spectrometer is highly efficient in resolving NIR signals. It can successfully distinguish two wavelengths at ~1557 nm and ~1558 nm. With larger (coarse) learning steps, the peak-to-peak-wavelength accuracy would be compromised, mainly due to a lower peak signal-to-noise ratio[11,14], as shown in Supplementary Fig. S5. We also conduct spectrometer stability and reproducibility measurements (Supplementary Fig. S6), which confirm operational stability over a ~10-day period and good device-to-device variation (~1.2 nm in the visible range and ~0.02 nm in the NIR range).

Figure 5 compares our results with state-of-the-art miniaturized spectrometers. Our single-junction tunnel diode-based spectrometer offers comparably high peak-wavelength accuracy with a broad operational window spanning the visible and NIR regions. Note that the operation bandwidth of our device is expected to cover ~4 μm, as the bandgap of multilayer BP is ~0.3 eV[27]. However, currently, we cannot demonstrate the operation in the mid-infrared range due to the unavailability of tunable light sources in the mid-infrared range for the learning and reconstruction process. In contrast to the previously reported triode-based spectrometers[12,14,15], where high gate voltage (typically >15 V) is required for operation, our tunnel diode-based spectrometer needs a lower voltage for tuning, which in principle reduces power consumption. Our diode-based spectrometer concept has the following main advantages: (1) simple and compact device structure without the gate terminal; (2) low driving voltage (<2 V) for

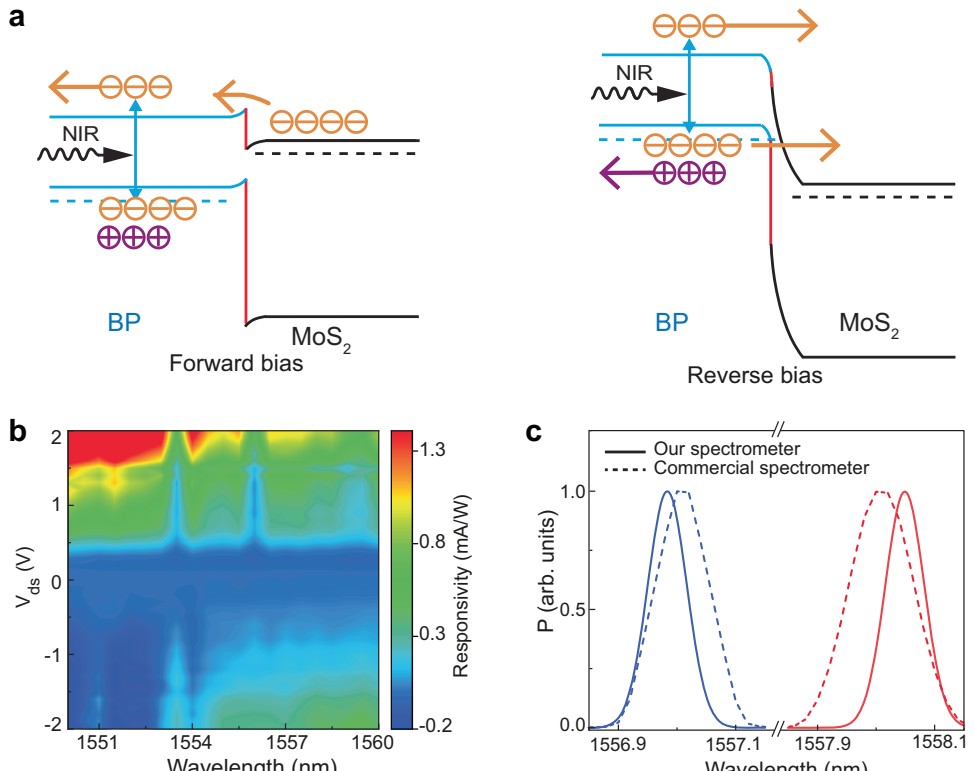

**Fig. 4 | Demonstration of our spectrometer at the NIR range. a** Schematic of photocurrent generation mechanism under the NIR light excitation under forward bias (left panel) and reverse bias (right panel) conditions. **b** Color contour plot of spectral response matrix obtained with a small learning step of -0.5 nm (21 different spectra, 41 voltage values per spectrum). **c** Reconstruction of typical quasi-monochromatic spectra with our spectrometer (solid curves). Dashed curves represent corresponding spectra measured with a commercial spectrometer.

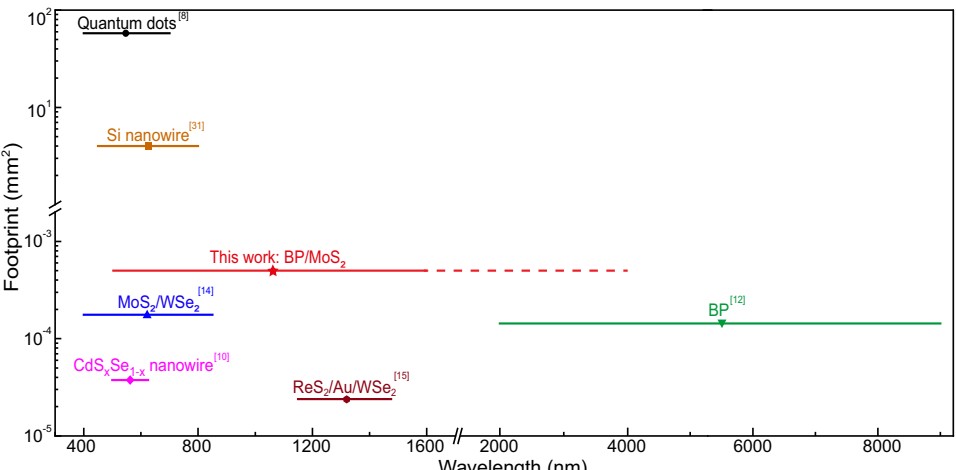

**Fig. 5 | Comparison of different reconstruction-type miniaturized spectrometers with bandgap engineering.** Systems are categorized according to their active materials. Solid lines in distinct colours mark corresponding operation bandwidth. The red dashed line suggests the theoretical extent of the operation bandwidth of our devices. Results are adopted from Refs. 8,10,12,14,15,31.

low power consumption; and 3) broadband operation from the visible to mid-IR for a much wider range of applications.

## Discussion

In conclusion, we have successfully demonstrated a reconstruction-type tunneling-diode-based broadband spectrometer. The spectrometer is electrically reconfigurable, and it does not require any filter or, photodetector array, or other bulky dispersive components to achieve high resolution with nanometer accuracy. The low operation voltage and compact footprint of the spectrometers offer immense possibility of integration with numerous portable applications such as consumer photonics and affordable on-chip spectral imaging (e.g., smart agriculture, remote sensing, and environmental monitoring).

*Note added in proof:* During proofreading, we became aware of recent work[28] showcasing a similar MoS2/BP heterojunction based

spectrometer operating in the ~1700 to 3600 nm range, validating our device's predicted mid-infrared functionality in Table 1.

## Methods

### Device fabrication and characterization

We used Si substrates with a 285 nm thick $SiO_2$ capping layer for device fabrication. hBN-encapsulated high-quality van der Waals heterostructures were prepared with commercially available hBN, BP, and 2H-phase $MoS_2$ crystals (2D Semiconductors, HQ Graphene) following the deterministic dry-transfer method. First, $BP/MoS_2$/bottom-hBN heterostructures were realized following a layer-by-layer flake stacking process that was performed with a motorized 2D transfer stage (Graphene Industries) placed inside an Argon gas-filled glovebox (MBRAUN) to maintain oxygen and water level below 0.1 ppm. Then, the chips were spin-coated with poly(methyl methacrylate) resist for patterning with electron beam lithography (EBL Raith EBPG5200). Afterwards, to realize 5/100 nm thick Ti/Au metal contacts on the EBL patterned structures, the chips were loaded to a vacuum electron beam evaporator (Angstrom) followed by acetone lift-off. Then, top hBN flakes were transferred onto the heterostructures inside a glovebox to protect them from atmospheric degradation. For process optimization, devices were annealed at 180 °C in a high vacuum (<0.05 mPa) for 2 h (Vacuum Furnace Webb). For additional protection against ambient conditions, an $Al_2O_3$ layer of thickness 20 nm was deposited at 120 °C employing an atomic layer deposition process (Beneq TFS-500). Finally, the Ti/Au electrodes were connected to a printed circuit board (PCB) by wire bonding (Wire Bonder Bondtec 5330). The topology of the fabricated hBN encapsulated $BP/MoS_2$ heterostructures was analyzed with an atomic force microscope (AFM Dimension Icon, Bruker).

### Raman measurements

All Raman spectra were acquired at room temperature using a micro-Raman spectrometer (WITec alpha300R) in a confocal backscattering geometry. A frequency-doubled Nd:YAG solid-state laser at 532 nm was used for excitation. The incident beam was focused perpendicularly onto the samples with a spot size of ~1 μm using an objective lens (100X, 0.9 NA). The backscattered signals were collected through the same objective lens and subsequently dispersed with an 1800-groove/mm grating and detected by a Si-charge-coupled camera. All experiments are performed at low excitation power ($P \leq 0.2$ mW) to avoid thermal damage to the samples.

### Electrical and optoelectrical measurements

Wire-bonded device chips were mounted to our home-built setup consisting of source meters (Keithley 2400 and 2401), and a custom-designed optical system allowed us to focus the laser beam on the desired locations of the spectrometer devices. Source and drain electrodes were connected to $MoS_2$ and BP, respectively. The drain to source voltage was always swept from −2 V to +2 V (which is well below the breakdown voltage of BP[29–30]). We used a supercontinuum laser source (SuperK Extreme, NKT Photonics) for photo-measurements in the visible range. The spectral bandwidth of the source was controlled by a tunable spectral filter (SuperK Varia, NKT Photonics) that allowed us to tune the spectral bandwidth to ~10 nm. For photo-measurements at the NIR region, we used a distributed-feedback laser source (Photonetics Osics 3610RA00). All measurements were done at room temperature, and data acquisition was performed with a customized LabVIEW program. For spectral references, the visible and the NIR spectra were measured by commercial spectrometers (FLAM-S and Anritsu MS9740A from OceanOptics).

## Data availability

All data needed to evaluate the findings of this study are available within the Article, its Supplementary Information and Source Data files and are also available from the corresponding author upon request. Source data are provided with this paper.

## Code availability

Code used for the spectral reconstruction is publicly available at Github (https://github.com/fonig/Reconstruction)[14].

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

## Acknowledgements
The authors acknowledge the funding from the Academy of Finland (314810, 333982, 336144, 352780, 352930, and 353364), the Academy of Finland Flagship Program (320167, PREIN), the EU H2020-MSCA-RISE-872049 (IPN-Bio), Business Finland (AGATE), the Jane and Aatos Erkko foundation, the Technology Industries of Finland centennial foundation (Future Makers 2022), and ERC (834742,101082183). This research was conducted at the Micronova, Nanofabrication Centre of Aalto University.

## Author contributions
Z.S. conceived of the idea and supervised the research. M.G.U. designed the experiment, fabricated spectrometer devices, and performed measurements on all the devices. S.D., A.M.S., F.A., X.C., and H.H.Y. assisted M.G.U. in optical and electrical characterization. S.D., L.W., N.F., and W.C. developed the reconstruction code. M.G.U., S.D., L.W., W.C., and Z.S. analyzed the data. A.C.L., Z.Y., H.L., H.H.Y., and T.H. commented on the experimental results and helped with the data analysis. M.G.U. wrote the manuscript. All authors participated in the scientific discussion and contributed to the writing of the manuscript.

## Competing interests
The authors declare no competing interests.
