## [Peer Review File · Nature Communications]

REVIEWER COMMENTS

Reviewer #1 (Remarks to the Author):

The manuscript entitled "Broadband miniaturized spectrometers with a van der Waals tunnel diode" by Zhipei Sun and co-workers presents a single device spectrometer, where the optical components of traditional spectrometers is replaced by cognitive operations based on the analysis of data.

In this work, the authors present a heterostructure of BP and MoS₂, extending the spectral range of previously studied TMDC-based heterojunction diodes.

The device shows a significant electrical tunable response, enabling the construction of visible-NIR spectrometer.

The topic is appealing and is of high interest to the broad readership of the journal.

However, I have the following questions and comments to the authors:

1. Properties of the response matrix and the procedure described in the paper:

a. To the best of your understanding, does the matrix represent a one-to-one mapping operator between the photocurrent and the spectrum ?

b. Have the authors tried to find the (pseudo) inverse of the responsively matrix (by SVD, RQ or any other algorithm)?

- what are the spectral properties of the matrix eigenvalues ?

- What is the performance of your spectrometer having applied the operation $P = \text{INV}(R) * I$, where INV is the inversion operator, compared to linear regression ?

c. What is the (training and testing) basis set size [how many different spectra (N) measured per voltage values (M), and how many Gaussians represents each spectral measurement] in construction of the response matrix R ?

2. Using Ridge regression with Gaussian basis set sometimes results in a subjective prediction of the a test data. For example, if the authors test the spectrometer with a reference spectrum vector (P) having more than one peak, or some general shape, it would demonstrate that R matrix predictions are free from such over fitting. One such case may be observed in Figure 3C green line and dash, but it is not clear to me if the dash follows the second small peak there. Please clarify.

3. Figure 4 demonstrates the spectral resolution of the device in the NIR.

- What is the size of the matrix in Figure 4b ?

- How is the resolution affected by an interpolation of the R matrix to resize by factors of 2,4, 8 in size ?

- Alternatively, what is the effect of using a smaller spectral range ?

Reviewer #2 (Remarks to the Author):

In the manuscript titled "Broadband Miniaturized Spectrometers with a Van der Waals Tunnel Diode," the authors present a highly capable broadband spectrometer achieved through the implementation of a van der Waals heterostructure diode. This innovation harnesses the distinct properties of chosen active van der Waals materials—specifically, molybdenum disulfide and black phosphorus. These materials exhibit electrically tunable photo-response, synergizing effectively with advanced computational algorithms engineered for spectral reconstruction. The manuscript is captivating and warrants further consideration for publication. To enhance its content, I propose a few minor revisions. Below are my comments:

1. The precision of bimodal reconstruction serves as a litmus test for the spectrometer's resolution. Could the authors elaborate on the limiting resolution when dealing with the reconstruction of two closely spaced narrowband peaks?

2. In the context of repeated measurements of identical light sources, it would be invaluable to elucidate the spectrometer's operational stability. This factor is pivotal in gauging the spectrometer's real-world applicability.

3. How accurate is the broadband reconstruction? It appears that the manuscript does not include a demonstration of the reconstruction for the wide spectrum.

4. What is the minimum detectable light intensity? In the case of a reconstruction spectrometer relying on response modulation, this aspect consistently presents a challenge.

5. To what extent does the spectrometer demonstrate robustness against measurement noise? How do external noise or measurement-related disturbances influence the overall performance of the spectrometer? Are there methods available to mitigate the adverse effects of measurement noise on reconstruction accuracy?

Reviewer #3 (Remarks to the Author):

Overall, the authors designed and fabricated a miniaturized spectrometer based on BP/MoS₂ heterostructure. Their van der Waals heterostructure diode switches the charge carrier transport from thermionic emission to band-to-band tunneling with a small source-drain bias voltage, which enables high wavelength resolution and broad bandwidth from vis to NIR. The present work is an interesting and novel follow-up to the authors' 2022 Science paper on a single-detector computational spectrometer using a MoS₂/WSe₂ heterojunction. This work is a substantial contribution in its own right as the authors have here demonstrated broadband operation with a simple device structure. Hence, I suggested the acceptance of this manuscript if the authors can address the comments/questions noted below:

1. The authors can provide more quantitative discussions on noise in the photodetector, such as noise-equivalent power or specific detectivity. Since BP has a narrow bandgap, charge carriers are thermally generated at very high rates at room temperature. Device performance cannot be discussed by responsivity alone, especially in the IR region.
2. Could the authors provide insight into how this concept can be expanded to a further wavelength regime longer than NIR?
3. Given that BP is relatively unstable than TMDs, how stable is this device's operation? Reproducibility and device-to-device variation can be provided Fig. 3b and Fig. 4b.

A point-by-point response letter to the reviewers' comments on the manuscript entitled "Broadband miniaturized spectrometers with a van der Waals tunnel diode" (*Research Article, No. NCOMMS-23-38160*)

Dear Editor and Reviewers,

We thank all three reviewers for their very positive comments on our manuscript. Indeed, Reviewer #1 comments that "*The topic is appealing and is of high interest to the broad readership of the journal...*"; Reviewer #2 evaluates that "*The manuscript is captivating and warrants further consideration for publication...*"; Reviewer #3 comments that "*This work is a substantial contribution... Hence, I suggested the acceptance of this manuscript...*".

The reviewers have also commented on aspects of our manuscript. In this letter, we provide point-by-point response to their comments. The sections with texts highlighted in **blue** indicate our responses, while the sections highlighted in **red** indicate the revisions in the main text or in the supplementary information.

Reviewer: 1

The manuscript entitled "Broadband miniaturized spectrometers with a van der Waals tunnel diode" by Zhipei Sun and co-workers presents a single device spectrometer, where the optical components of traditional spectrometers is replaced by cognitive operations based on the analysis of data. In this work, the authors present a heterostructure of BP and MoS₂, extending the spectral range of previously studied TMDC-based heterojunction diodes. The device shows a significant electrical tunable response, enabling the construction of visible-NIR spectrometer. The topic is appealing and is of high interest to the broad readership of the journal.

However, I have the following questions and comments to the authors:

We thank the reviewer for the very positive comments.

I. Properties of the response matrix and the procedure described in the paper:

a. To the best of your understanding, does the matrix represent a one-to-one mapping operator between the photocurrent and the spectrum ?

We thank the reviewer for the query.

Practically, the mapping may not be one-to-one due to two potential reasons. First, due to measurement noise, the measured photocurrent of the same input spectrum could differ from time to time. Second, when the reconstruction problem is under-determined, different input spectra can lead to the same set of photocurrent signals, indicating the existence of multiple solutions. Thus, it is necessary to include *a priori* information during the reconstruction process. For example, in this work, we developed adaptive Tikhonov reconstruction algorithm to fully utilize the smoothness condition of the sought spectrum.

b. Have the authors tried to find the (pseudo) inverse of the responsively matrix (by SVD, RQ or any other algorithm)?

— *what are the spectral properties of the matrix eigenvalues ?*

— *What is the performance of your spectrometer having applied the operation $P = INV(R) * I$, where INV is the inversion operator, compared to linear regression ?*

We thank the reviewer for the valuable comments. Yes, we had tried SVD to find the pseudo inverse of the response matrix.

We agree with the reviewer that it is helpful to check the spectral properties of the matrix eigenvalues/singular values of the response matrix. The spectrum of the singular values is calculated and plotted in Fig. R1 of this reply letter. As can be seen, the singular values gradually decay to zero.

Figure R1: The spectral properties of the matrix singular values.

This spectrum property implies that the problem to be solved is ill-posed and effectively underdetermined with small singular values that are sensitive to noise and error, which often appears in the field of reconstructive spectrometers and other linear inverse problems.

As suggested, we compared the performance of pseudo-inverse with linear regression (our method) in Fig. R2. The reconstruction of the pseudo-inverse operator is dominated by oscillations with large amplitude, as shown in Fig. R2(a). As mentioned above, due to the ill-posedness of the problem, the solution is severely damaged by measurement noise. In comparison, our adaptive Tikhonov regularization method with non-negative constraints (U. Kurokawa *et al.*, IEEE Sens. J., **11**, 1556 (2011)), a typical linear regression algorithm, can stabilize the solution through regularization, making it more adept at addressing ill-posed inverse problems (Fig. R2(b)).

Figure R2: The reconstruction results of a spectrum with a 600 nm central peak from (a) pseudo-inverse, and (b) our algorithm (adapted from Fig. 3c of the revised manuscript).

c. What is the (training and testing) basis set size [how many different spectra (N) measured per voltage values (M), and how many Gaussians represents each spectral measurement] in construction of the response matrix R ?

We thank the reviewer for the questions.

The basis set size in the training step (Fig. 3b of the revised manuscript), during construction of the response matrix R for the visible (500-800 nm) range, is 31×41 (31 different spectra, 41 voltage values per spectrum). For testing, 151 Gaussians represent each spectral measurement.

The basis set size in the training step (Fig. 4b of the revised manuscript), during construction of the response matrix R for the NIR (1550-1560 nm) range, is 21×41 (21 different spectra, 41 voltage values per spectrum). For testing, 2000 Gaussians are used for the high-resolution NIR reconstruction.

To address the reviewer's comments, we have now updated the caption of Fig. 3(b) on Page 7 of the revised manuscript as follows: “Color contour plot of experimentally obtained spectral response matrix (31 different spectra, 41 voltage values per spectrum).”

Similarly, we have updated the caption of Fig. 4(b) on Page 8 of the revised manuscript as: “Color contour plot of spectral response matrix obtained with a small learning step of ~0.5 nm (21 different spectra, 41 voltage values per spectrum).”

2. Using Ridge regression with Gaussian basis set sometimes results in a subjective prediction of the a test data. For example, if the authors test the spectrometer with a reference spectrum vector (P) having more than one peak, or some general shape, it would demonstrate that R matrix predictions are free from such over fitting. One such case may be observed in Figure 3C green line and dash, but it is not clear to me if the dash follows the second small peak there. Please clarify.

We thank the reviewer for the comment.

Indeed, reconstruction of the complex spectrum can be a more complicated task. The second peak in Figure 3C is a computational error which can arise in similar models. In principle, such artifacts can be suppressed by filtering or improving the base model.

To address the reviewer's comments, we have now updated the main text on Page 8, paragraph 1 of the revised manuscript as follows: “The small second peak at ~675 nm is a computational error. In principle, such artifacts can be suppressed by filtering or improving the base model.”

3. Figure 4 demonstrates the spectral resolution of the device in the NIR.

— *What is the size of the matrix in Figure 4b ?*

— *How is the resolution affected by an interpolation of the R matrix to resize by factors of 2,4, 8 in size ?*

— *Alternatively, what is the effect of using a smaller spectral range ?*

We thank the reviewer for the questions.

Figure 4b of the revised manuscript corresponds to a response matrix $R(N, M)$ of size 21×41 (i.e., 21 different spectra, 41 voltage values per spectrum).

We calculate the wavelength peak position corresponding to the 1557 nm wavelength reconstruction. As shown in Fig. R3, we do not observe any significant effect on the resolution when the \mathbf{R} matrix is enlarged by factors of 2, 4, and 8 (i.e., 2, 4 and 8 times finer spectral resolutions). This is because the original \mathbf{R} matrix is already densely measured.

Figure R3: Effect of interpolation of the responsivity matrix on the peak wavelength accuracy of our spectrometer when the matrix size is enlarged.

This observation is consistent with our previous work (H. H. Yoon *et al.*, *Science*, **378**, 296 (2022)), where we report that the resolution improvement saturates after a certain resize factor, as presented in Fig. R4.

Figure R4: Peak signal-to-noise ratio (between reconstructed and reference spectra) as a function of the learning step. Adopted from H. H. Yoon *et al.*, *Science*, **378**, 296 (2022).

Reviewer: 2

In the manuscript titled "Broadband Miniaturized Spectrometers with a Van der Waals Tunnel Diode," the authors present a highly capable broadband spectrometer achieved through the implementation of a van der Waals heterostructure diode. This innovation harnesses the distinct properties of chosen active van der Waals materials—specifically, molybdenum disulfide and black phosphorus. These materials exhibit

electrically tunable photo-response, synergizing effectively with advanced computational algorithms engineered for spectral reconstruction. The manuscript is captivating and warrants further consideration for publication. To enhance its content, I propose a few minor revisions. Below are my comments:

We thank the reviewer for the very positive comments.

1. The precision of bimodal reconstruction serves as a litmus test for the spectrometer's resolution. Could the authors elaborate on the limiting resolution when dealing with the reconstruction of two closely spaced narrowband peaks?

We thank the reviewer for the valuable suggestion.

There are three factors that limit the resolution of the spectrometers. (1) Step size during construction of the learning matrix. A smaller step size in the learning matrix typically enhances the resolution of the spectrometer. (2) Contrast of photo-response of the device with respect to the bias voltages and the wavelengths. Photo-response corresponding to each bias voltage should ideally show a strong wavelength dependence. Indeed, this is the premise for reconstructive algorithms to distinguish the wavelengths. (3) The spectrometer's resolution is also limited by measurement noise. When the differences in photo-response at adjacent wavelengths are dominated by noise rather than the photo-response itself, dual peaks at those positions are hard to distinguish.

For example, we explored the resolution limit of our device using the dense response matrix with a 0.5 nm learning step (Fig. 4b of the revised manuscript) via numerical simulation. Given that the measurement noise may damage the feature of response at different wavelengths itself, which may degrade the device's resolution, we tested the bimodal reconstruction performance under three groups of different Gaussian random noise levels, as shown in Fig. R5 of the reply letter. In a 2% noise level simulation environment, the device can resolve a dual peak of 2 nm (Fig. R5a-b), and lower noise levels lead to higher peak height accuracy (Fig. R5d e g h). Further, a dual peak of 1 nm can be successfully distinguished when the noise level is reduced to 0.5% (Fig. R5i).

Figure R5: Simulated reconstruction of bimodal spectra with our algorithm in different noisy conditions.

2. In the context of repeated measurements of identical light sources, it would be invaluable to elucidate the spectrometer's operational stability. This factor is pivotal in gauging the spectrometer's real-world applicability.

We thank the reviewer for the insightful comment.

As suggested by the reviewer, we carried out operational stability measurements with identical light sources. From the repetitive measurements, as shown in Fig. R6 of the reply letter, it is obvious that the device is quite stable with insignificant deviation (~ 0.03 nm with ~ 10 days) among the peak position of reconstructed spectra (colored solid lines) compared to that of the reference spectrum (dashed line in black) measured with a commercial spectrometer. We also update the results as SI figure S5(b) of the revised SI.

Figure R6 (SI figure S5(b)): Operational stability of the spectrometer under repeated measurements of identical light sources.

To address the reviewer comments, we have now updated the main text on Page 9, paragraph 1 of the revised manuscript as follows: “We also conduct spectrometer stability and reproducibility measurements (Supplementary Fig. S5), which confirm good operational stability over a ~10-day period ...”. We also added the results in the Supplementary information (Supplementary Fig. S5).

3. How accurate is the broadband reconstruction? It appears that the manuscript does not include a demonstration of the reconstruction for the wide spectrum.

We thank the reviewer for raising the issue.

We performed measurements for broadband reconstruction, as the reviewer suggested. Figure R7 shows the performance of our spectrometer in broadband spectra reconstruction. The black and red solid lines represent the spectra reconstructed with our spectrometer and measured using a commercial spectrometer (FLAM-S, Ocean Optics), respectively. The results show that our spectrometer agrees well with a commercial spectrometer regarding wide spectral measurement.

Figure R7 (SI figure S3): Broadband spectrum measurement comparison between our spectrometer and a commercial spectrometer (FLAM-S, Ocean Optics). The full-width half maximum of the spectrum is ~ 180 nm.

To address the reviewer's comments, we have now updated the main text on Page 8, paragraph 1 of the revised manuscript as follows: "We also carry out broadband spectrum measurement (Supplementary Fig. S3), which agrees well with that of the commercial spectrometer." We also added the results in the Supplementary information (Supplementary Fig. S3).

4. *What is the minimum detectable light intensity? In the case of a reconstruction spectrometer relying on response modulation, this aspect consistently presents a challenge.*

We thank the reviewer for the comment.

Our MoS₂/BP van der Waals tunnel diode is very sensitive to optical excitation. As shown in Fig. R8, it can detect optical power as low as 100 nW.

Figure R8: Responsivity of our MoS₂/BP tunnel diode under different optical powers.

5. *To what extent does the spectrometer demonstrate robustness against measurement noise? How do external noise or measurement-related disturbances influence the overall performance of the spectrometer? Are there methods available to mitigate the adverse effects of measurement noise on reconstruction accuracy?*

We thank the reviewer for the comments.

We have already used regularization methods in our reconstruction (Details on page 4 of the supplementary Information) to improve the robustness against measurement noise. In response to your question 1, we

discussed the effect of noise on the reconstruction of bimodal spectra. Here, we discuss the effect of noise on the reconstruction of broadband spectra (Fig. R9 of this reply letter). The results demonstrate the high robustness of our spectrometer against measurement noise for broadband spectrum experiments due to the nature of our adaptive Tikhonov regularization method. For question 2, as mentioned above, due to the influence of measurement noise, the feature of response at different wavelengths may be damaged, and the measured photocurrent of the same input spectrum may differ from time to time. This could result in a larger deviation in the reconstructed spectrum compared to the ground truth. Indeed, there are other methods to mitigate the adverse effects of the measurement noise except our regularization methods, for example, employing more advanced denoising methods, e.g., denoising autoencoder (J. Zhang *et al.*, IEEE Sens. J., **21**, 6450 (2021)), which can reduce noise in the raw measurements before performing the reconstruction.

Figure R9: Simulated reconstruction of the broadband spectrum with our algorithm in different noisy conditions.

Reviewer: 3

Overall, the authors designed and fabricated a miniaturized spectrometer based on BP/MoS2 heterostructure. Their van der Waals heterostructure diode switches the charge carrier transport from thermionic emission to band-to-band tunneling with a small source-drain bias voltage, which enables high wavelength resolution and broad bandwidth from vis to NIR. The present work is an interesting and novel

follow-up to the authors' 2022 Science paper on a single-detector computational spectrometer using a MoS₂/WSe₂ heterojunction. This work is a substantial contribution in its own right as the authors have here demonstrated broadband operation with a simple device structure. Hence, I suggested the acceptance of this manuscript if the authors can address the comments/questions noted below:

We thank the reviewer for the positive comments and valuable suggestions.

1. The authors can provide more quantitative discussions on noise in the photodetector, such as noise-equivalent power or specific detectivity. Since BP has a narrow bandgap, charge carriers are thermally generated at very high rates at room temperature. Device performance cannot be discussed by responsivity alone, especially in the IR region.

We thank the reviewer for the comment.

We thank the reviewer for the suggestion. We employed BP/MoS₂ heterojunctions as a tunneling diode, so the charge carrier transport is not entirely determined by a narrow bandgap of BP. The thermally emitted carriers will contribute to the dark current, while the photocurrent is mainly determined by interlayer carrier transport (photoemission and recombination) across the heterointerface. Following the reviewer's comment, we performed noise-equivalent power (NEP) calculations on the device presented in Fig. 3 and Fig. 4 of the revised manuscript. As shown in Fig. R10 of this reply letter, NEP of the device is on the order of ~nanowatt per square root of hertz, which is comparable to that of the previously reported BP photodetectors (e.g., L. Huang *et al.*, ACS Appl. Mater. Interfaces, **9**, 36130 (2017)).

Figure R10: Noise equivalent power (NEP) of our MoS₂/BP tunnel diode under visible and NIR light illumination.

2. Could the authors provide insight into how this concept can be expanded to a further wavelength regime longer than NIR?

We thank the reviewer for the comment.

Earlier studies report BP photodetectors (e.g., Q. Guo *et al.*, *Nano Lett.*, **16**, 7, 4648 (2016); X. Chen *et al.*, *Nat. Commun.*, **8**, 1672 (2017); S. Yuan *et al.*, *Nat. Photonics*, **15**, 601 (2021)), focusing on mid-infrared light detection (2-9 μm) due to that the bandgap of relatively thick BP is at the mid-infrared range. Therefore, in principle, our BP/MoS₂ heterostructure can work at longer wavelengths (e.g., the mid-infrared range). However, we could not assess the performance of our spectrometers in the mid-infrared range due to the limitation of our experimental setup (i.e., missing a tunable mid-infrared laser for training & testing).

3. Given that BP is relatively unstable than TMDs, how stable is this device's operation? Reproducibility and device-to-device variation can be provided Fig. 3b and Fig. 4b.

We thank the reviewer for the comment.

Indeed, the intrinsic instability of mono- and few-layer nanosheets of BP (of thickness <10 nm) poses a challenge in BP-based device applications (G. Abellán *et al.*, *JACS*, **139**, 10432 (2017)), which can be improved by various passivation methods. For example, a thin Al₂O₃ (~5 nm) hBN coating layer protects BP flakes, exfoliated on SiO₂ substrates, from degradation in ambient conditions for ~6-8 months (e.g., Y. Illarionov *et al.*, *ACS Nano*, **10**, 9543 (2016); Gamage *et al.*, *Nanotechnology*, **28**, 265201 (2017)). Our spectrometer device uses multilayer BP flakes (~50 nm thick). Further, we encapsulate the BP/MoS₂ heterostructure with (top and bottom) hBN, where an additional ~20 nm thick ALD Al₂O₃ coating layer is deposited on the top hBN flake. This ensures reliable passivation and stability of our devices.

Figure R11(a) of the reply letter compares the current density of a freshly prepared BP/MoS₂ tunnel diode with that of the same device after around one month. The excellent stability of the device refers to high-quality passivation of the BP/MoS₂ heterostructure. Further, our spectrometers show operational stability under repeated measurements of identical light sources (Fig. R11(b)). Figure R11(c-d) of the reply letter benchmarks the performance of our best two spectrometers against the commercial spectrometers. The device-to-device variation is reasonable, such as ~1.2 nm in the visible range (Fig. R11(c)), ~0.02 nm in the NIR range (Fig. R11(d)).

Figure R11 (SI figure S5): Operational stability and device-to-device variation of our spectrometer. (a) Comparison of dark current density of a device over a month. (b) Operational stability of our spectrometer under repeated measurements of identical light sources. (c-d) Comparison of spectral reconstruction performance of our best two devices working in the visible (c) and the NIR (d) ranges.

To address the reviewer comments, we have now updated the main text on Page 9, paragraph 1 of the revised manuscript as follows: “We also conduct spectrometer stability and reproducibility measurements (Supplementary Fig. S5), which confirm good operational stability over a ~10 day period and good device-to-device variation (~1.2 nm in the visible range and ~0.02 nm in the NIR range).”

We thank all reviewers for their valuable comments, which have significantly improved our manuscript. We hope now the revised manuscript can be accepted for publication.

The authors.

REVIEWERS' COMMENTS

Reviewer #1 (Remarks to the Author):

I would like to thank the authors for their point by point response, answering all the questions and comments in a clear and satisfying manner.

The manuscript contains a valuable contribution to the field of computational spectrometers and should be accepted for publication in Nature Communications.

I would like to ask the authors to include as much information from our comments and responses in the supplementary information, such that it would serve the readers who are trying to make their first steps into the field.

Reviewer #2 (Remarks to the Author):

The authors have satisfactorily addressed the reviewers' concerns. I believe this manuscript can be now accepted in the present form.

Reviewer #3 (Remarks to the Author):

In the revised manuscript, the authors have addressed all my concerns. I recommend the publication of the manuscript.

A *point-by-point response letter* to the reviewers' comments on the manuscript entitled "Broadband miniaturized spectrometers with a van der Waals tunnel diode" (Research Article, No. NCOMMS-23-38160)

Dear Editor and Reviewers,

We thank all three reviewers for their very positive comments on our manuscript and for recommending it for publication. In this letter, we provide a point-by-point response to their comments. The sections with texts highlighted in blue indicate our responses.

Reviewer: 1

I would like to thank the authors for their point by point response, answering all the questions and comments in a clear and satisfying manner. The manuscript contains a valuable contribution to the field of computational spectrometers and should be accepted for publication in Nature Communications.

I would like to ask the authors to include as much information from our comments and responses in the supplementary information, such that it would serve the readers who are trying to make their first steps into the field.

We thank the reviewer for the very positive comments and for recommending the manuscript for publication.

We provided broadband reconstruction, operation stability, and reproducibility data of the spectrometer in the supplementary information. In addition, we are now adding the power-dependent responsivity and the noise-equivalent-power (NEP) characterization results of the spectrometer devices (Fig. S3 in the SI).

Reviewer: 2

The authors have satisfactorily addressed the reviewers' concerns. I believe this manuscript can be now accepted in the present form.

We thank the reviewer for the very positive comments and for recommending the manuscript for publication.

Reviewer: 3

In the revised manuscript, the authors have addressed all my concerns. I recommend the publication of the manuscript.

We thank the reviewer for the very positive comments and for recommending the manuscript for publication.

We thank all reviewers for their valuable comments, which have significantly improved our manuscript. We hope now the revised manuscript can be accepted for publication.

Yours sincerely,

The authors